# Effects of Neighboring Units on the Estimation of Particle Penetration Factor in a Modeled Indoor Environment

**Yonghang Lai [1], Ian A. Ridley [1] and Peter Brimblecombe [1,2,3,4,*]**

[1] School of Energy and Environment, City University of Hong Kong, Hong Kong; yonghalai2-c@my.cityu.edu.hk (Y.L.); ian.alex.ridley@gmail.com (I.A.R.)
[2] Department of Marine Environment and Engineering, National Sun Yat-Sen University, Kaohsiung 804, Taiwan
[3] Aerosol Science Research Center, National Sun Yat-Sen University, Kaohsiung 804, Taiwan
[4] School of Environmental Sciences, University of East Anglia, Norwich NR4 7TJ, UK
[*] Correspondence: p.brimblecombe@uea.ac.uk

**Abstract:** Ingress of air from neighboring apartments is an important source of fine particulate matter ($PM_{2.5}$) in residential multi-story buildings. It affects the measurement and estimation of particle deposition rate and penetration factor. A blower-door method to measure the particle deposition rate and penetration factor has previously been found to be more precise than the traditional decay-rebound method as it reduces variability of $PM_{2.5}$ ingress from outside. CONTAM is a multi-zone indoor air quality and ventilation analysis computer program to aid the prediction of indoor air quality. It was used in this study to model the indoor $PM_{2.5}$ concentrations in an apartment under varying $PM_{2.5}$ emission from neighboring apartments and window opening and closing regimes. The variation of indoor $PM_{2.5}$ concentration was also modeled for different days to account for typical outdoor variations. The calibrated CONTAM model aimed to simulate environments found during measurement of particle penetration factor, thus identifying the source of error in the estimates. Results show that during simulated measurement of particle penetration factors using the blower-door method for three-hour periods under a constant 4 Pa pressure difference, the indoor $PM_{2.5}$ concentration increases significantly due to $PM_{2.5}$ generated from adjacent apartments, having the potential to cause an error of more than 20% in the estimated value of particle penetration factor. The error tends to be lower if the measuring time is extended. Simulated measurement of the decay-rebound method showed that more $PM_{2.5}$ can penetrate inside if the $PM_{2.5}$ was generated from apartments below under naturally variable weather conditions. A multiple blower-door fan can be used to reduce the effects of neighboring emission and increase the precision of the penetration estimates.

**Keywords:** air change rate; blower door; CONTAM model; Hong Kong; $PM_{2.5}$

## 1. Introduction

Poor indoor air quality can affect human health. As multi-apartment dwellings become an increasing part of urban life, pollution within them assumes greater importance. In contemporary cities residents generally spend 80–90% of their time indoors [1,2]. Apart from the indoor sources, the penetration of outdoor airborne contaminants through ventilation and infiltration is also one of the most important factors in controlling air quality indoors [3]. Thus, in multistory residential buildings in larger cities, airborne contaminants may arise not only from the outside environment, but additionally through inter-zone transmission [4]. This provides another important pathway for indoor air pollutants and health impacts [5]. An important example was the outbreak of severe acute respiratory syndrome, or SARS, in 2003 [6,7]. Here cross-infection contributed to the spread of the virus, which was found in rooms where residents did not suffer from the disease [8], particularly for single-sided naturally ventilated buildings in Hong Kong. In multistory

buildings, people tend to close the windows and doors to lessen the risk of disease and airborne contaminant transmission, especially in winter. However, transmission remains possible, both vertically and horizontally, through adjacent apartment or lift areas due to stack and wind effects; this being driven by three forces: the buoyancy force, wind force, and their combination [5]. Several previous studies investigated pollutant transmission in inter-unit buildings: Liu et al. [9] numerically investigated the airborne transmission in a high-rise building with buoyancy-driven ventilation. Wu and Niu [10] show that the airflow through the window was weakened under a buoyancy-driven mechanical exhaust system. Wang et al. [11] used a Computational Fluid Dynamics (CFD) model to investigate pollutant transmission inside and outside a multistory building with single-sided, buoyancy-driven ventilation, and dependence on window type. Mu et al. [12,13] used a wind tunnel test to study the inter-apartment gaseous pollutant dispersion and transmission vertically and horizontally in a rectangular multistory building, indicating the pollutant dispersion routes are strongly influenced by wind direction and source location. In addition, the combination of the buoyancy and wind forces may more often be seen in the real case. For example, Mao et al. [14] applied CONTAM and CFD models to study the transmission routes of airborne pollutants in a 33-story residential building and revealed the range of factors that control pollutant spread in both horizontal and vertical directions. Not surprisingly there are large reductions (3−4 orders of magnitude) in contaminant concentrations when rooms are widely separated. Window opening is a key factor that influences indoor air quality, the effects of window opening in adjacent units is therefore also likely to be important.

Recently, some studies also investigated air infiltration and pollution transmission using a combination of CONTAM with and without CFD models. Wind pressure, as a function of wind speed, direction, building configuration, and local terrain effects, can be accounted for by CONTAM with one of three options: constant wind pressure; surface average wind pressure profile and spatially and time-varying wind pressure profile [15]. The external link of CFD model can be coupled with CONTAM to consider the effects of variable external wind pressures in a non-cubical surface. For instance, Argyropoulos et al. [16] applied CONTAM model with a CFD module to predict the indoor and outdoor particulate matter (PM) building infiltration in an office building in Qatar, under normal conditions and severe dust storm event. The prediction presented an encouraging agreement with measurement when the Heating Ventilation and Air Conditioning (HVAC) system was used, and attributed this to accurate estimation of wind pressure and representation of building envelope. However, when considering the whole-building and yearly dynamic simulations, modeling building air infiltration, and computational speed, the multi-zone CONTAM model without CFD might be better [15]. Underhill et al. [17] applied the CONTAM and EnergyPlus to simulate annual energy consumption and yearly average indoor PM concentration for 648 building scenarios. Zhu et al. [18] also investigated the ventilation rates in different ventilation conditions in college residential halls by using multi-zone CONTAM model. The model was calibrated by $CO_2$ decay measurement, and strategies for reducing the risk of transmission of respiratory infection were proposed. Thus, the multi-zone CONTAM model is also associated with indoor air quality model.

When the indoor particle deposition rate is estimated based on an indoor air quality model, measurements of indoor and outdoor concentrations are required. In our previous work [19], the blower-door method was used to estimate the particle deposition rate and penetration factor under steady-state pressure differences. Higher precision for this method was found when the experimental methods were repeated compared to the traditional decay-rebound method. However, an error was still observed in a multi-story building. Since our previous work used only the indoor and outdoor concentrations in estimates, the transmission from neighboring zones was likely to affect indoor concentration, through activities such as cooking in adjacent units. The particles generated by neighbors may easily transfer to the test apartment when the blower-door method is used, so the estimates of particle penetration factor readily influenced due to the variation of indoor concentrations.

The effect of internal sources in neighboring dwellings on the changes of indoor particle concentration is not well studied, despite its potential influence when estimating particle penetration factors in both blower-door and decay-rebound methods.

In this study, we made observations in an apartment of a multistory building; first, estimating the PM$_{2.5}$ deposition rate and penetration factor by measuring indoor and outdoor concentrations using both the blower-door and decay-rebound methods. CONTAM was used as a model to simulate air change rate and the concentration of indoor PM$_{2.5}$ in the test apartment and three adjacent units. Results from the model were compared and validated with the measurements. Moreover, the inter-transmission of PM$_{2.5}$ within neighboring units was then studied: the effects of PM$_{2.5}$ emission, as a model for cooking, from adjacent units vertically and horizontally on the indoor PM$_{2.5}$ concentration were investigated. The changes in PM$_{2.5}$ accumulation and the coefficient of variation of estimated particle penetration factors was explored along with the effect of opening windows of adjacent units. An enhanced blower-door method is proposed for determining the proportional air change rate of different areas based on an indoor air quality model. The analytical results were compared with the simulated output from the CONTAM model.

## 2. Materials and Methods

### 2.1. Building Description

The experimental measurements used to validate the CONTAM simulations were conducted in an eight-story residential building (Figure 1). This building is located on the campus of City University of Hong Kong (CityU) in Tat Chee Avenue, Kowloon, Hong Kong. One apartment on the fifth floor, height ~15.6 m above the ground, was selected as the test apartment and adjacent units that are directly (test apartments on upper and lower floors) or indirectly (opposite adjacent apartment) connected with the test unit horizontally and vertically, as shown in Figure 1b. The test apartment consisted of two bedrooms, kitchen, bathroom and living areas, with a total floor area of 77.6 m$^2$ and a volume of 201.8 m$^3$. The external walls were made from concrete, with exterior ceramic sidings and an internal finish of vinyl covered drywall. The doors of the corridor connecting the stairway were normally closed as the door is used as an emergency exit. The lift door was also closed during the test to reduce the stack effect in the building. The balcony areas were regarded to be the outside for the purpose of these experiments. This apartment has three 1.1 kW air conditioners, located in the living room and two bedrooms. There are two exhaust fans located in the kitchen and bathroom. The leakage areas for the whole apartment were determined using blower-door pressurization tests with a measurement uncertainty of around 5% [20]. These values were used in the CONTAM model. Although different residential buildings have different floor layouts, the layout of this test building represents a basic form for residential apartments in Hong Kong, including two apartments connected with one common area (corridor, lift, and stairway) on the same floor. It provides a good example for showing the changes of indoor PM$_{2.5}$ concentration in multistory buildings.

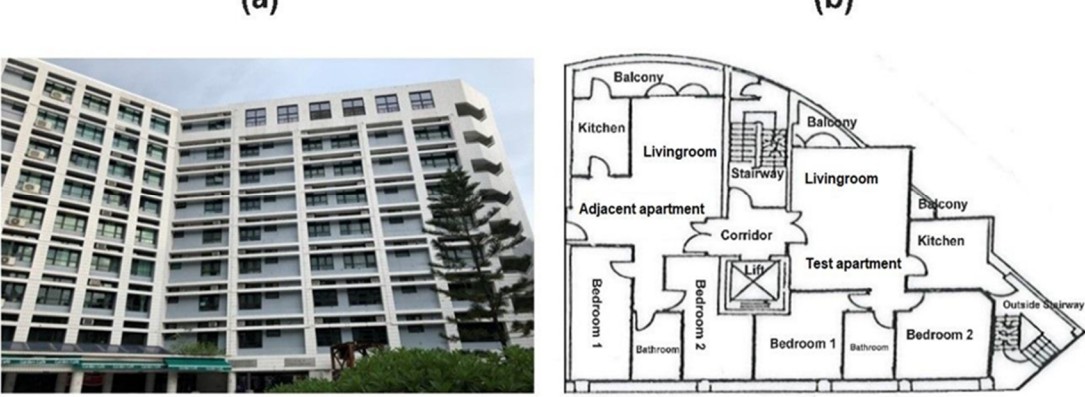

**Figure 1.** (**a**) The multistory apartment building; (**b**) floor layout; identical on each floor.

## 2.2. Field Measurements

The field measurements were made to determine transmission of $PM_{2.5}$, by measuring indoor and outdoor $PM_{2.5}$ concentrations under the constant indoor-outdoor pressure difference of 4 Pa (imposed by the blower door) and natural conditions. Each location was measured on four different days, between 8 September and 16 September 2017. Table 1 summarizes the test information over the whole experimental period, and the indoor temperatures were measured continuously for the nine days. Since 4 Pa is widely used as reference pressure in many standards (e.g., American Society for Testing and Materials ASTM, American Society of Heating, Refrigerating and Air-Conditioning Engineers, ASHRAE) to define the Effective Leakage Area (ELA), here it has been taken to representt a weather induced pressure [21]. During the tests, all windows and external doors were closed, while all internal doors were open and there were no pollutant sources. The envelope airtightness was measured using the Retrotec Blower Door System (Retrotec, Everson, WA USA) and established the air infiltration rate. The airflow rate at different pressure differences can be calculated by the power-law equation:

$$Q_f = C\,\Delta P_f{}^n \tag{1}$$

where $Q_f$ is the air flow ($m^3\,h^{-1}$), $C$ is the flow coefficient ($m^3\,h^{-1}\,Pa^{-n}$), $\Delta P$ is the indoor-outdoor pressure difference (Pa), $n$ is the pressure exponent and the subscript f relates to the fan-induced flow or pressure, with $C$ and $n$ determined by least squares fitting.

**Table 1.** Test information: setting; average indoor and outdoor temperature; overall air change rate; wind speed. Note: SD is the standard deviation.

| Setting | Test Date | Indoor T (SD[a]) °C | | | Air Change Rate h$^{-1}$ | Outdoor Conditions | |
|---|---|---|---|---|---|---|---|
| | | Living Room | Bedroom 1 | Bedroom 2 | | T (SD) °C | W (SD) m s$^{-1}$ |
| **Blower-door fan** | 8 September 2017 | 30.36 (0.32) | 30.27 (0.27) | 30.28 (0.30) | 0.71 | 27.58 (4.46) | 1.41 (1.69) |
| | 9 September 2017 | 30.69 (0.15) | 30.53 (0.15) | 30.57 (0.17) | 0.69 | 28.59 (2.00) | 1.33 (1.10) |
| | 15 September 2017 | 31.77 (0.26) | 31.60 (0.22) | 30.64 (0.13) | 0.70 | 26.94 (3.54) | 0.83 (0.72) |
| | 16 September 2017 | 32.01 (0.34) | 31.83 (0.32) | 30.79 (0.10) | 0.71 | 26.89 (2.42) | 1.32 (0.84) |
| **Natural condition** | 11 September 2017 | 31.36 (0.36) | 31.19 (0.34) | 30.22 (0.27) | 0.23 | 27.79 (4.48) | 0.77 (0.88) |
| | 12 September 2017 | 31.78 (0.19) | 31.65 (0.22) | 30.71 (0.16) | 0.29 | 27.50 (5.46) | 0.95 (0.81) |
| | 13 September 2017 | 31.82 (0.42) | 31.71 (0.30) | 30.90 (0.20) | 0.12 | 25.85 (5.56) | 0.49 (0.88) |
| | 15 September 2017 | 31.77 (0.26) | 31.60 (0.22) | 30.64 (0.13) | 0.25 | 26.94 (3.54) | 0.83 (0.72) |

Three $PM_{2.5}$ concentration monitors (DUSTTRAK 8530EP, TSI, Shoreview, MN, USA) along with $CO_2$ loggers (HOBO MX $CO_2$ logger, Onset, Bourne, MA, USA) were installed in the center of the three main rooms: living room, bedroom 1 and 2. The three monitors were inter-calibrated by measuring the concentration in the same place over the same period, as shown in Figure 2. A further monitor was installed on the balcony representing the outdoor concentration, along with temperature and relative humidity loggers (HOBO U12, Onset, Bourne, MA, USA). One anemometer (HOBO T-DCI-F900-L-O, Onset, Bourne, MA, USA) was installed on the external wall, on the side with the blower-door system to measure the wind speed. All monitors measured at a 1-min logging interval. The indoor data were carefully calibrated when relative humidity (RH) was higher than 60% [22]. Gravimetric calibration using DustTrak's internal filter cassette with Teflon filter was conducted before the tests, weighing each three times on an electronic microbalance in the laboratory following calibration guidelines. The deposition rates and penetration factors for $PM_{2.5}$ were determined by measuring the indoor and outdoor concentrations under the blower-door fan, and natural ventilation. The blower-door method adopted is as detailed in a previous study [19,23]: a constant indoor-outdoor pressure difference of seven pressure differences (4, 6, 8, 10, 12, 14, and 16 Pa) was created, and the indoor and outdoor concentrations were recorded simultaneously. A single time constant ($1/k_{ACH}$) was adopted as the time required to reach the steady-state conditions to estimate the deposition

rate and penetration factor based on an indoor air quality model [19]. The overall air change rates were obtained from the blower-door test. Under natural conditions, the widely used $CO_2$ decay and rebound method [24–26] was adopted: after injecting $CO_2$ inside, the air change rate was estimated from the decrease of $CO_2$ concentrations over two hours. A high-pressure fan was then used to pressurize the indoor air to outside to achieve a low concentration of $PM_{2.5}$, three hour rebound periods were adopted to estimate the penetration factor as they provide a relatively long time to reach an approximately steady condition [24]. During the test, no people remained in the test or neighboring apartments except the experimentalist who controlled the blower-door devices.

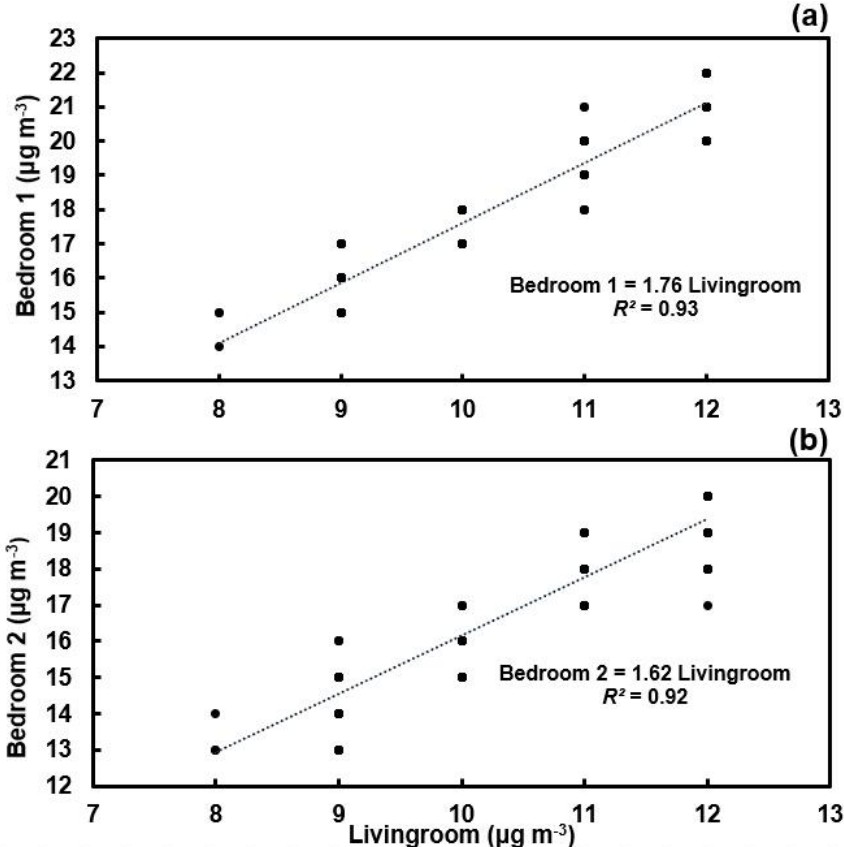

**Figure 2.** Calibration between fine particulate matter ($PM_{2.5}$) monitor in living room and bedroom 1 (**a**) and in living room and bedroom 2 (**b**).

*2.3. Mass Balance Model*

Estimates of deposition rate and penetration factor for the test apartment, considered outdoor concentrations (Equation (2)), and the average indoor concentrations of three zones (living room, bedroom 1 and bedroom 2) and assumed the interior well-mixed without indoor sources:

$$\frac{dc_i}{dt} = Pk_{ACH,\Delta p}c_o - \left(k_{ACH,\Delta p} + v_{\Delta p}\right)c_i \tag{2}$$

where $c_i$ and $c_o$ are the average indoor and outdoor particle concentrations, respectively; $P$ is the penetration factor; $k_{ACH,\Delta p}$ is the air exchange rate at $\Delta p$; $v_{\Delta p}$ is the deposition rate at $\Delta p$. When the $\frac{dc_i}{dt}$ is not zero, the indoor particle concentration can be solved by a backward differential scheme for a given time step, as shown below:

$$c_i(t) = Pk_{ACH,\Delta p}c_o(t-1)\Delta t + \left(1 - \left(k_{ACH,\Delta p} + v_{\Delta p}\right)\Delta t\right)c_i(t-1) \tag{3}$$

where $\Delta t$ is the time step, which is consistent with the logging interval (1 min). The deposition rate was estimated by assuming the $P$ as unity as the blower-door fan is large

enough for all particles to be blown inside during pressurization sets [19]. Thus, penetration factor (*P*) was calculated by minimizing the sum of squared errors based on air change rate ($k_{ACH,\Delta p}$), deposition loss rate (*k*) and indoor and outdoor concentrations.

### 2.4. Multi-Zone Simulation

Multi-zone airflow and contaminant transport modeling have been widely used to investigate the transport of outdoor particles indoors and have been able to show differences between measurement and CONTAM simulation. For example, Hu et al. [27] simulated particle resuspension in a three-zone office building, modeling indoor particle deposition and resuspension. Rim et al. [28] studied the entry of size-resolved ultrafine particles into a test building, finding that deposition and penetration influenced particle transmission. Dols et al. [29] investigated indoor and outdoor dynamics of fine particles in a two-story office building showing the importance of proper parameterization for predicting airflow and transport of particles using CONTAM.

In the present study, the CONTAM version 3.2 (NIST, Gaithersburg, MD, USA) was used to simulate the time-transient air change rate and $PM_{2.5}$ concentration of two apartments in the 8-floor building under two conditions: 4 Pa depressurization set and natural conditions. Figure 3 illustrates a test apartment with the adjacent unit at the same floor in the CONTAM graphical interface, which depicts different zones, airflow paths (doors, wall joints, windows, etc.) and a simple Air Handling System with supply and return flow. Building exterior leakage data were obtained from a blower-door test, with a total flow coefficient (*C*) is 0.0345 $m^3\ s^{-1}\ Pa^{-n}$ and a pressure exponent (*n*) 0.599. Considering the stack effect, the leakage of each envelope on individual floors was divided into three heights on each wall, representing the lower relative elevation of 0.1 m, middle relative elevation of 1.3 m, and upper relative elevation of 2.5 m. The leakage of the floor and ceiling are at zero relative elevation. The leakage data were applied to all exterior walls, ceilings, and floors, under one-way flow using the power law function in CONTAM following the CONTAM User Guide [30]. Large openings with height (*H*) of 2 m, width (*W*) of 0.8 m and a default discharge coefficient of 0.78 were used to represent the single internal door opening [31]; a one opening two-way flow model was used.

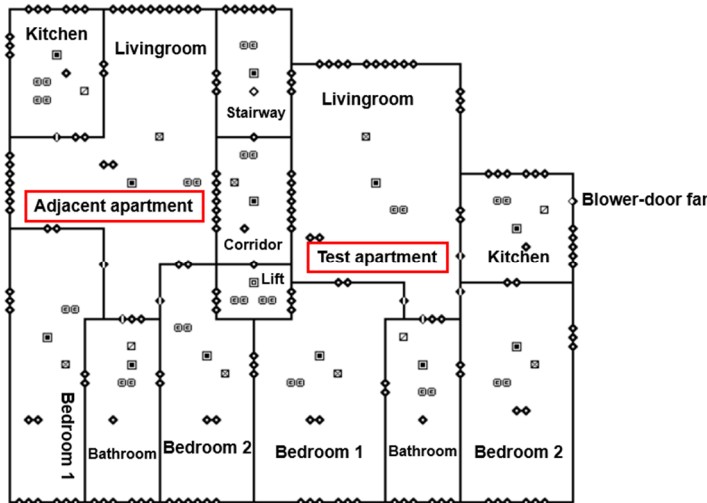

**Figure 3.** Graphic interface for each floor of the multistory building, where the blower-door fan was only installed in the test apartment.

The wind effects of each flow path were computed using an average wind pressure profile based on wind pressure coefficient ($C_{wind}$) relationships found in Swami and Chandra [32]. When the blower-door method was used, the effects of interior partition should be negligible and the whole building average infiltration was estimated. The direction of external air pressure can be controlled due to the steady-state indoor-outdoor pressure

difference for reducing the effects of variable external wind pressure, and the estimates of deposition rate and penetration factor were calculated by indoor air quality model. Thus, the average wind pressure profile for CONTAM based on multi-zone might be better considering the blower-door method. A wind speed modifier of 0.36 corresponding to suburban terrain [30] was used for all outdoor leakage paths. These wind pressure data were used in CONTAM to account for the effects of local terrain on the variation of wind speed with height above ground level [33]. The measured indoor temperatures were set as a schedule for each zone. Weather for the typical meteorological year (TMY) was used to represent the transient weather data, which were obtained from the website of EnergyPlus weather files (Hong Kong 450,070 (CityUHK). Since the location of the apartment is on the campus of CityU, we used the monitoring sites at the university, so measurements were made close to the test building. The weather information during the experimental days is given in Table 1.

Table 2 shows estimated deposition rates and penetration factors for the test periods. These estimates were based on the Equation (3) that only considers the outdoor concentration without people in neighboring units. The average deposition rate for $PM_{2.5}$ was computed to be $0.13\,h^{-1}$ and $0.12\,h^{-1}$ under natural pressure differences for the blower-door method and $CO_2$ decay-rebound method, respectively. Although the estimated deposition rate is lower than experimental results often reported in the literature [34–36], a similar result was found by Long et al. [37]; that there was better agreement between their data and the theoretical predictions. The average penetration factor was calculated to be 0.89 based on the 4 Pa depressurization set, while 0.84 for $CO_2$ decay-rebound method. The blower-door method gives higher precision for the estimates with lower coefficient of variation as shown in Table 2, which shows a good agreement with a previous study [23].

**Table 2.** Deposition rate (*v*) and penetration factor (*P*) of four replicate tests with the coefficient of variation (*Cv*) percentage.

| Method | Test Date | Deposition Loss Rate (*v*) h$^{-1}$ | Penetration Factor (*P*) |
|---|---|---|---|
| **Blower-door** | 8 September 2017 | 0.14 | 0.89 |
| | 9 September 2017 | 0.12 | 0.86 |
| | 15 September 2017 | 0.12 | 0.91 |
| | 16 September 2017 | 0.13 | 0.90 |
| | Average (*Cv*) | 0.13 (8%) | 0.89 (2%) |
| **Decay-rebound** | 11 September 2017 | 0.14 | 0.85 |
| | 12 September 2017 | 0.01 | 0.81 |
| | 13 September 2017 | 0.18 | 0.87 |
| | 15 September 2017 | 0.14 | 0.83 |
| | Average (*Cv*) | 0.12 (63%) | 0.84 (3%) |

In most residential apartments in multi-rise buildings, the external doors are usually closed, but not the windows. When the windows are open, the airflow rate tends to be larger and the penetration factor of $PM_{2.5}$ approaches unity. Then the variations of indoor $PM_{2.5}$ concentrations tend to be similar with that outdoors. The test apartment has five casement windows (one in the bathroom, kitchen, living room, and the two bedrooms). The windows are of different sizes; those in the unmonitored kitchen and bathroom are smaller. The one-way flow calculated from the power law model with the orifice area data formula was used to simulate the window openings, and the cross-sectional area of all windows in the test apartment was measured and input to the model: $34,000\,cm^2$ for living room, $11,150\,cm^2$ for kitchen, $8125\,cm^2$ for bathroom, and $26,350\,cm^2$ for bedroom 1 and 2. Other parameters adopted the default set including a transition Reynolds number of 30, discharge coefficient of 0.6, and flow exponent of 0.5 [30].

When the windows are closed, the ELA depends on the gaps and cracks of the window frames along with those from external walls or joints [38]. In this case, the leakage area per

unit length of window-frame gaps adopts the reference value in the ASHRAE handbook, i.e., 0.24 cm$^2$ m$^{-1}$. The total length of the window-frame gaps was measured as 19.6 m for the living room, 8 m for kitchen, 6.9 m for bathroom and 17.4 m for each bedroom. The reference pressure 4 Pa was used with the discharge coefficient of 1 and a flow exponent of 0.599. Since the floor surface in the test apartment is made from wood, it assumed an ELA of 0.19 cm$^2$ m$^{-2}$ from the ASHRAE value for the parquet flooring. The ELA of ceiling was set to 0.82 cm$^2$ for each surface-mounted light. These values all formed input to the model for calculating the air flow.

On the other hand, the blower-door test was also applied to establish the air tightness and the exterior leakage, and the one-way power-law model was used to simulate the air flow rate based on the assigned flow coefficient (*C*) and pressure exponent (*n*). As the blower-door test can only measure the total flow coefficient and pressure exponent of the whole apartment, the numbers of each air flow path should be determined. Assuming the flow coefficient of each air flow path is the same, the sum of flow coefficients should be the same as the blower-door test result, i.e., 0.0345 m$^3$ s$^{-1}$ Pa$^{-n}$, with the same pressure exponent (*n*) of 0.599. In this study, a 2.5 multiplier for flow elements was used in CONTAM, which represents the numbers of cracks in the building envelope. There is no strict rule for determining the multiplier for air flow paths, but the total air flow rate calculated from blower-door tests should be the same as those calculated from ASHRAE leakage data, including floor, ceiling, and window gaps etc. It suggests that the power law model, with the orifice area data (window) has good agreement with the one-way power-law model with only flow coefficient and exponent (only cracks) that was calibrated using measurements. The total number of the gaps can then be found as it depends on the ratio of the total flow coefficient and the numbers of gaps. In our experimental apartment, a total 144.5 gaps showed the best agreement with those calculated based on the reference value in the ASHRAE handbook, so that the flow coefficient (*C*) of each airflow path was calculated to be 0.000239 m$^3$ s$^{-1}$ Pa$^{-n}$.

### 2.5. Statistical Analysis

The validation of predictions used measured particle concentrations adopting ASTM D5157 Standard Guide for Statistical Evaluation of Indoor Air Quality Models [39]. Three parameters were used: correlation coefficient (*R*), regression slope (*M*), and regression intercept (*b*); perfect prediction represented as a slope of 1.0, intercept of 0.0 and coefficient of 1.0. Three other parameters including the error and bias within ASTM D5157 are normalized mean square error (NMSE), fractional bias (FB), and fractional bias of variance (FS):

$$NMSE = \frac{\overline{\left(C_p - C_o\right)^2}}{\overline{C_p}\,\overline{C_o}} \tag{4}$$

$$FB = 2\frac{\left(\overline{C_p} - \overline{C_o}\right)}{\left(\overline{C_p} + \overline{C_o}\right)} \tag{5}$$

$$FS = 2\frac{\left(\delta^2_{C_p} - \delta^2_{C_o}\right)}{\left(\delta^2_{C_p} + \delta^2_{C_o}\right)} \tag{6}$$

where $c_p$ and $c_o$ are the simulated and experimental PM$_{2.5}$ concentrations, respectively, and $\delta^2$ is the variance (averaged as denoted by bar). When normalized mean square error (NMSE) is zero, a perfect agreement is expected, and a higher value indicates a larger difference between measurement and prediction. Zero values for fractional bias (FB) and fractional bias based on variations (FS) suggests a perfect agreement between measurement and prediction. An acceptable agreement between measurements and predictions should meet the following six criteria:

(1)　The correlation coefficient (*R*) $\geq$ 0.9;

(2)   The regression line between measurements and simulations should have a slope (*M*)
      between 0.75 and 1.25;
(3)   An intercept (b) should less than 25% of the average measured concentration;
(4)   The normalized mean square error (NMSE) $\leq 0.25$;
(5)   Absolute values of normalized or fractional bias (FB) $\leq 0.25$;
(6)   Absolute value of fractional bias based on the variance (FS) $\leq 0.50$.

## 3. Results and Discussion

### 3.1. Indoor PM$_{2.5}$ Concentration: Simulation and Measurements

Figure 4a–h present the PM$_{2.5}$ concentrations from CONTAM simulations versus measurements from the living room with statistical analysis and measured concentrations in three indoor zones (living room, bedroom 1, and bedroom 2) and outside over a duration related to the air change rate of the room [23] under the 4 Pa depressurization set on different days. The relationships between simulated and measured concentrations are shown in Figure 4a,c,e,g) on 8, 9, 15, and 16 September, respectively. The figures indicate a good agreement between measured and predicted PM$_{2.5}$ concentrations from different days, although the variations of fractional bias (*FB*) of days are seen. There are differences in PM$_{2.5}$ change among the three zones (Figure 4b–h), so PM$_{2.5}$ concentrations in bedroom 1 were generally a little lower than that of other zones (Figure 4b), although this was true every day, hinting at the influence of external variations, including weather and outdoor PM$_{2.5}$ concentrations. Figure 5a–h further compares CONTAM results and measurements and the 3-h observations from the rooms and outdoors under natural conditions, on 11 (a, b), 12 (c, d), 13 (e, f), and 15 (g, h) September. The figures show poorer fits with lower R$^2$, compared with the constant pressure set of Figure 4. It is likely to be caused by increased propagation of variable outdoor conditions (e.g., wind speed, indoor–outdoor temperature gradient) under natural ventilation. All data were calibrated using real weather data by using on-site measurement.

Although the statistical analyses show that the agreement between observation and model prediction is not perfect for all cases, it is encouraging to note that most statistical parameters are in the acceptable range, meeting the six criteria of ASTM D5157 Standard Guide. In general, most measured PM$_{2.5}$ concentrations of the three indoor zones show similar positive trends in depressurization and natural conditions, thus, it is reasonable to use the average values for estimating the overall deposition rates and penetration factors. As the modeled PM$_{2.5}$ concentrations of both sets in different days illustrate an acceptable agreement with the measurement values, the predicted concentrations are well trusted and can be extended to model the variations of PM$_{2.5}$ concentrations over longer times.

### 3.2. Simulating Variability

The coefficient of variation of indoor concentration across four days in different seasons was investigated, by simulating the 24-h indoor PM$_{2.5}$ variations under −4 Pa indoor-outdoor pressure difference induced by a blower-door fan and natural conditions in CONTAM. The four months were chosen to include March, June, September, and December representing four seasons and the middle day of each of the months (15th) was selected. All other parameters including deposition rate, penetration factor, flow coefficients, etc., were kept the same.

Figure 6 illustrates the 24-h coefficient of variation for concentration across the single days drawn from each season, and a low and stable coefficient is seen when a constant pressure difference was created over the whole period. While under natural conditions, a large variation is observed, up to ~60% for three zones, six times higher than when the blower-door (modeled as a fan) was in place. The changes of ambient conditions also influence the indoor concentration of neighboring apartments, which further affected the test apartment. Thus, the measurement variation across four seasons can be reduced using the blower-door method in a multistory apartment.

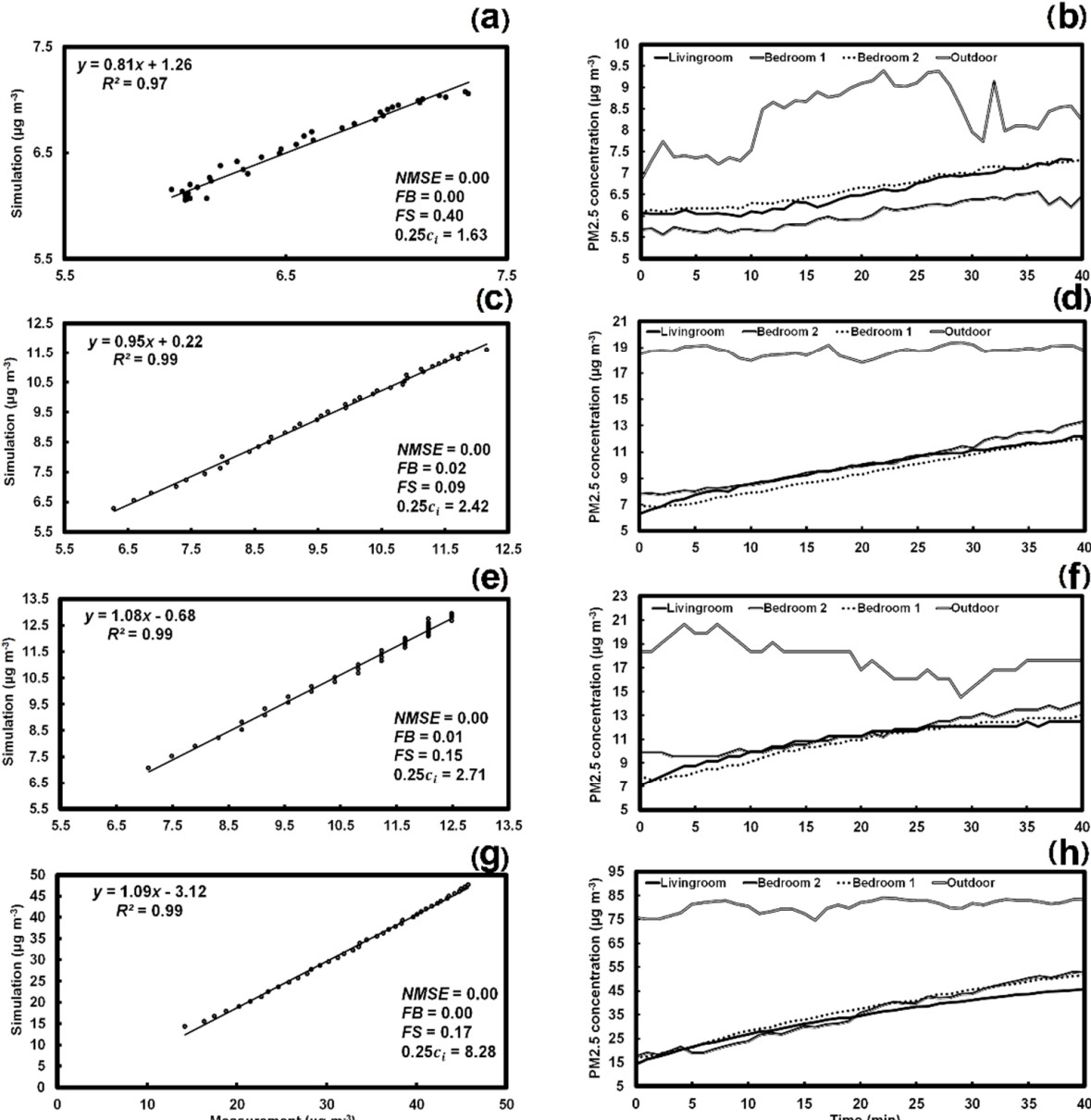

**Figure 4.** PM$_{2.5}$ concentration of CONTAM simulation and measurement in the living room (**a**,**c**,**e**,**g**) and the measured PM$_{2.5}$ concentration (**b**,**d**,**f**,**h**) in the living room and bedrooms 1 and 2, and outside under the 4 Pa depressurization set on four different days: 8 September (**a**,**b**), 9 September (**c**,**d**), 15 September (**e**,**f**), and 16 September (**g**,**h**). The y is the simulation, and x is the measurement.

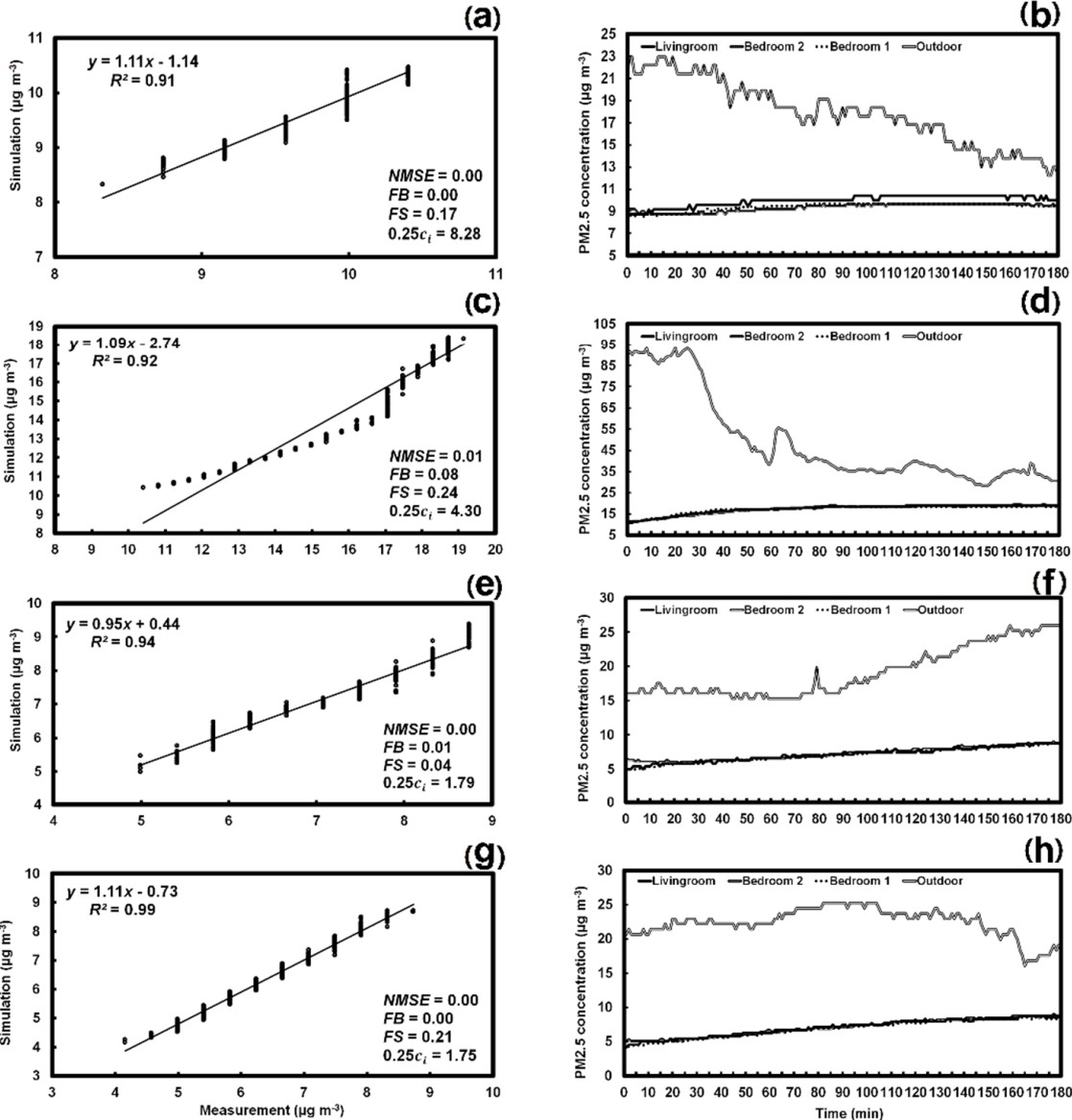

**Figure 5.** PM$_{2.5}$ concentration of CONTAM simulation and measurement in the living room and the measured PM$_{2.5}$ concentration in four places (living room, bedroom 1, bedroom 2, and outside) under three-hour natural conditions on four different days: 11 September (**a**,**b**), 12 September (**c**,**d**), 13 September (**e**,**f**), and 15 September (**g**,**h**), respectively. The y is the simulation, and x is the measurement.

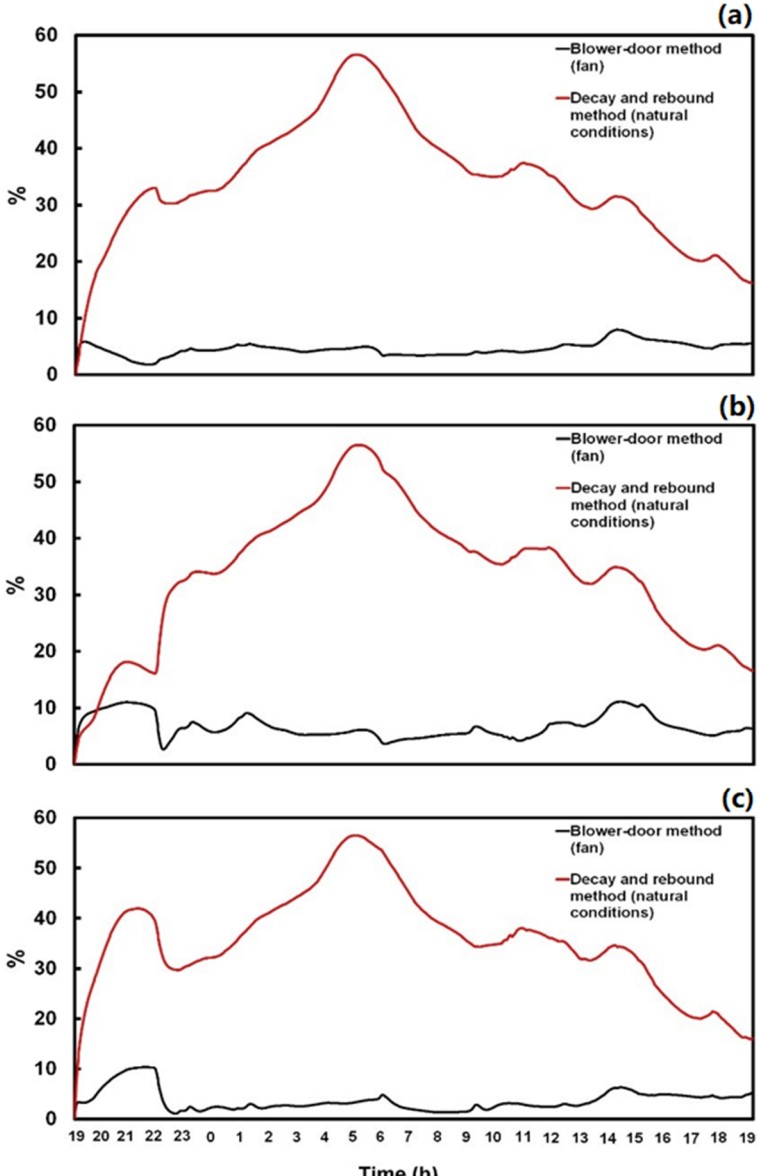

**Figure 6.** Coefficient of variation (%) of indoor PM$_{2.5}$ concentration in the living room (**a**), bedroom 1 (**b**) and bedroom 2 (**c**) under two conditions (−4 Pa pressure difference induced by fan and natural conditions) across four days in different seasons.

### 3.3. Effect of Emissions from Adjacent Apartments

Three-hour and 24-h indoor PM$_{2.5}$ concentrations over four different periods (8–9; 9–10; 12–13; 15–16 September) were modeled using CONTAM, by changing the emission rates of the three adjacent apartments: (i) opposite neighboring apartments on the same floor, (ii) upper floor, and (iii) lower floor apartments. The four days were selected to cover a relatively wide range of wind speeds as seen in Table 1: 0.49 to 1.41 m s$^{-1}$. Integration of the PM$_{2.5}$ concentration over time was used to represent an accumulated PM$_{2.5}$ level. The kitchen was set as the location of PM$_{2.5}$ generation. One-hour periods of particle generation with five constant generation rates: 0.4, 0.8, 1.2, 1.6, and 2.0 mg min$^{-1}$ were used; realistic for PM$_{2.5}$ emission rate for different types of cooking [40]. Since outdoor weather conditions are one of the most important influence on indoor particle exchange or accumulation [19,41], outdoor concentrations were not changed when the indoor concentrations were modeled for different days. The simulation of 15 September was selected as an ideal case because measurements and simulations were well correlated ($R^2$ = 0.99) under both conditions for all three zones. The 24-h indoor PM$_{2.5}$ concentrations from 15–16 September

were modeled and the accumulation on different days presented. The penetration factors estimated from indoor and outdoor concentrations during two periods (first three-hour and 24-h) under blower-door conditions were computed based on Equation (3). The error in the estimated penetration factor under different emission rates was compared with those of the no-generation condition, with all other parameters kept the same in the CONTAM model. It reveals the effects of neighboring emission on estimates from the blower-door method.

### 3.3.1. PM$_{2.5}$ Accumulation, Error of Penetration Factor and Window Closure

Window opening is one of the most important controls on indoor pollutant levels [28], but there are few investigations of the effect of windows in adjacent apartments. Figure 7 illustrates the increased percentages of the PM$_{2.5}$ accumulating in the three indoor zones at five emission rates in the three adjacent apartments: (a) when all windows were closed or (b) open in opposite neighboring apartments on the same day. Higher generation rates always result in increased percentages of indoor PM$_{2.5}$ level for the whole apartment. The increased percentages found when windows are closed are much higher than those for all zones, when windows are open. The highest value is up to 20% for closed windows, but only 0.07% for open windows. It suggests that the indoor PM$_{2.5}$ concentration is less influenced by the PM$_{2.5}$ generation of neighboring apartments when all windows are open, probably because open windows result in larger air change rates and mean that the increased particle concentration in neighboring apartments disperse outdoors. The resulting PM$_{2.5}$ concentrations tend to be similar even when generation rates change.

Figure 7a further shows the percentage increase in PM$_{2.5}$ concentrations under two conditions (constant 4 Pa and natural conditions). At 4 Pa, the largest percentage increase is seen in bedroom 1 when the PM$_{2.5}$ was generated in the adjacent apartment on the same floor, <20%. Regarding the layout of this building, bedroom 1 is connected directly to the adjacent zones, but the living room is not, revealing the corridor area provides more time for PM$_{2.5}$ to dilute and disperse. It further reduces the accumulation in the indoor areas, i.e., there are more air flow paths between test zones and corridor areas, as shown in Figure 3. The larger number of air flow paths, between indoor and adjacent apartments on the same floor, suggests that PM$_{2.5}$ would allow easier penetration indoors under a constant indoor-outdoor pressure difference. The lowest increase is seen in bedroom 2, probably because bedroom 2 is not directly or indirectly linked to any adjacent areas and is thus more easily influenced by outdoor variations.

When the blower-door fan was removed, yet PM$_{2.5}$ emission rates maintained as in the previous case, concentrations were almost unchanged. However, the greatest increase was observed across all indoor zones when generation sources were located in the apartment on the lower floor, even though there are fewer air flow paths between the two zones, so it is likely to be caused by stack effects. Opening adjacent windows did not cause a strong change of PM$_{2.5}$ concentrations, the largest, though small, increase still arises from the lower apartment (<0.07%, as shown in Figure 7b).

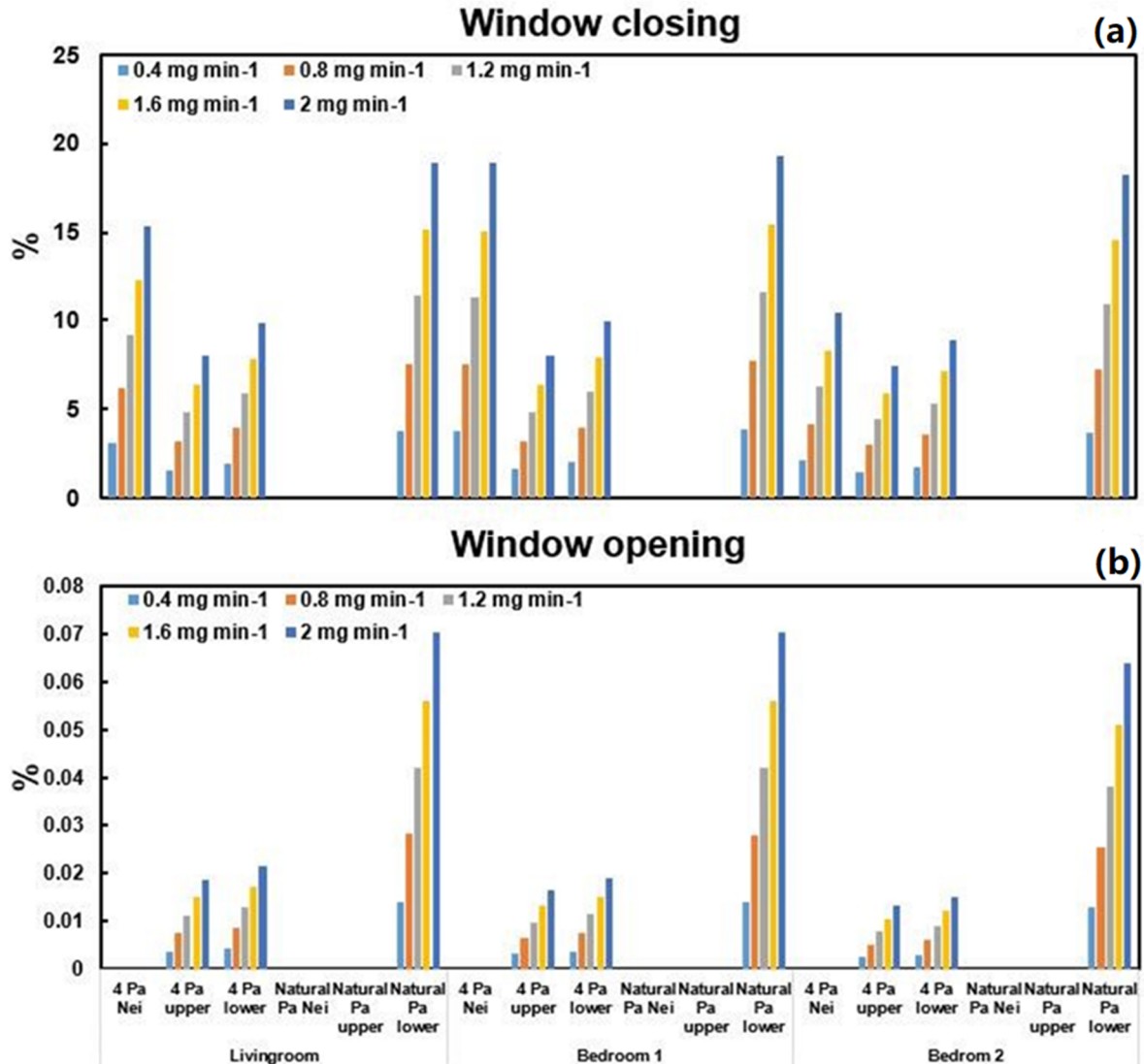

**Figure 7.** The percentage increase in PM$_{2.5}$ accumulation in the three indoor zones at five generation rates (0.4; 0.8; 1.2; 1.6; 2 mg min$^{-1}$) from three adjacent apartments: Opposite neighboring (Nei), Upper and Lower apartments, under two conditions (4 Pa depressurization and natural conditions), with windows closed (**a**) and opened (**b**) in neighboring apartments.

Table 3 lists the penetration factor estimated from Equation (3) based on the average of three indoor zones and outdoor PM$_{2.5}$ concentrations during two periods (first three-hour and 24 h) under 4 Pa pressure difference on the same day. The penetration factors tend to be similar, with the input value of 0.89 in CONTAM driven from field experiment when there was no emission from neighboring units across all conditions. However, an obvious difference in the estimates can be seen for different emission rates during the two periods. Since the emission of PM$_{2.5}$ will cease after an hour, representing an actual cooking period, the three-hour estimates cover a short-time duration, similar to that of the experimental duration, while 24-h embraces a longer time. With windows closed, the penetration factors increase at higher emission rates, and the values are always higher than unity for the three-hour period. The highest average penetration factor for all emission rates is 1.39, when the sources are in the adjacent apartment, where $Cv > 28\%$ for the three-hour period. The estimates for the sources coming from upper and lower apartments are only slightly less than those from the adjacent unit, with $Cv > 20\%$. It suggests that the estimated penetration factor has a large variation for different neighboring emission rates over short-time durations. However, the estimates tend to be similar when the testing duration extends to 24 h. All penetration factors are close to 0.89, similar to that with no

generation, and a *Cv* ~ 2% for different emission rates with sources present in the adjacent unit. When the window is open, the estimates of penetration factor are not influenced by neighboring emissions.

**Table 3.** Penetration factor estimated by modeling indoor concentration and outdoor value under 4 Pa pressure difference with five generation rates during two periods. *Cv* is the coefficient of variation of the mean.

| Time | Sources Location | Neighboring Window Condition | Emission Rate (mg min$^{-1}$) | | | | | | | |
|---|---|---|---|---|---|---|---|---|---|---|
| | | | **0** | **0.4** | **0.8** | **1.2** | **1.6** | **2.0** | **Mean** | ***Cv* (%)** |
| Three hours | Adjacent apartment | Closing | 0.86 | 1.07 | 1.28 | 1.49 | 1.70 | 1.91 | 1.39 | 28.37 |
| | | Opening | 0.91 | 0.91 | 0.91 | 0.91 | 0.91 | 0.91 | 0.91 | 0 |
| | Upper apartment | Closing | 0.86 | 1.00 | 1.14 | 1.28 | 1.42 | 1.56 | 1.21 | 21.65 |
| | | Opening | 0.89 | 0.89 | 0.89 | 0.89 | 0.89 | 0.89 | 0.89 | 0 |
| | Lower apartment | Closing | 0.86 | 1.02 | 1.19 | 1.36 | 1.53 | 1.69 | 1.28 | 24.53 |
| | | Opening | 0.89 | 0.89 | 0.89 | 0.89 | 0.89 | 0.89 | 0.89 | 0 |
| | Mean | | 0.88 | 0.96 | 1.05 | 1.14 | 1.22 | 1.31 | | |
| | *Cv* (%) | | 2.43 | 7.98 | 16.57 | 23.88 | 30.15 | 35.52 | | |
| 24 h | Adjacent apartment | Closing | 0.87 | 0.88 | 0.89 | 0.9 | 0.91 | 0.92 | 0.90 | 2.09 |
| | | Opening | 0.95 | 0.95 | 0.95 | 0.95 | 0.95 | 0.95 | 0.95 | 0 |
| | Upper apartment | Closing | 0.87 | 0.87 | 0.88 | 0.88 | 0.89 | 0.89 | 0.88 | 1.02 |
| | | Opening | 0.91 | 0.91 | 0.91 | 0.91 | 0.91 | 0.91 | 0.91 | 0 |
| | Lower apartment | Closing | 0.87 | 0.88 | 0.88 | 0.89 | 0.89 | 0.9 | 0.89 | 1.19 |
| | | Opening | 0.92 | 0.92 | 0.92 | 0.92 | 0.92 | 0.92 | 0.92 | 0 |
| | Mean | | 0.90 | 0.90 | 0.91 | 0.91 | 0.91 | 0.92 | | |
| | *Cv* (%) | | 3.75 | 3.39 | 3.03 | 2.73 | 2.44 | 2.27 | | |

The study also reveals that large variation in estimates when the neighboring windows are open or not during the three-hour period. Variation tends to be more obvious at higher emission rates, but the window status does not influence the 24-h duration, with all *Cv* > 4%. Compared with the three-hour estimates, a 24-h period test provides a more precise and accurate value. The precision and accuracy of the blower-door method could improve estimates of penetration factors because the emitted particles have longer to disperse outdoors or deposit in neighboring interiors.

3.3.2. PM$_{2.5}$ Accumulation and the Error of Penetration Factor on Different Days

Figure 8 shows the 24-h percentage increase of PM$_{2.5}$ concentration in (a) the living room, (b) bedroom 1 and (c) bedroom 2 on four days, when the sources in adjacent apartments over three levels were changed. The most dramatic increase in PM$_{2.5}$ on different days is seen when the PM$_{2.5}$ was produced on the lower floor. The highest percentage (>45%) for the three zones, is observed when the lower floor emission rate is 2 mg min$^{-1}$ (9 September) under natural conditions. The smallest increase from the source below is seen on 12 September, <7%, at 2 mg min$^{-1}$ under the 4 Pa set. At 4 Pa depressurization, the highest increase is observed when the sources come from the apartment on the same floor across all generation rates. This may result from the large gaps or cracks between adjacent apartments in multistory buildings (e.g., interior doors and windows), compared with the gaps between the ceiling and floor. Under natural conditions, when the sources are on the lower floor apartment, the percentage increase is much larger than the cases where sources are associated with other floors. It suggests that stack effects could be a dominating factor influencing indoor PM$_{2.5}$ where sources are below. The assumptions made in assigning the

distribution and location of the air gaps will of course influence the relative importance of different transport mechanisms. The accumulation in bedroom 2 tends to be small when the source is from opposite apartments as bedroom 2 is not directly attached to it.

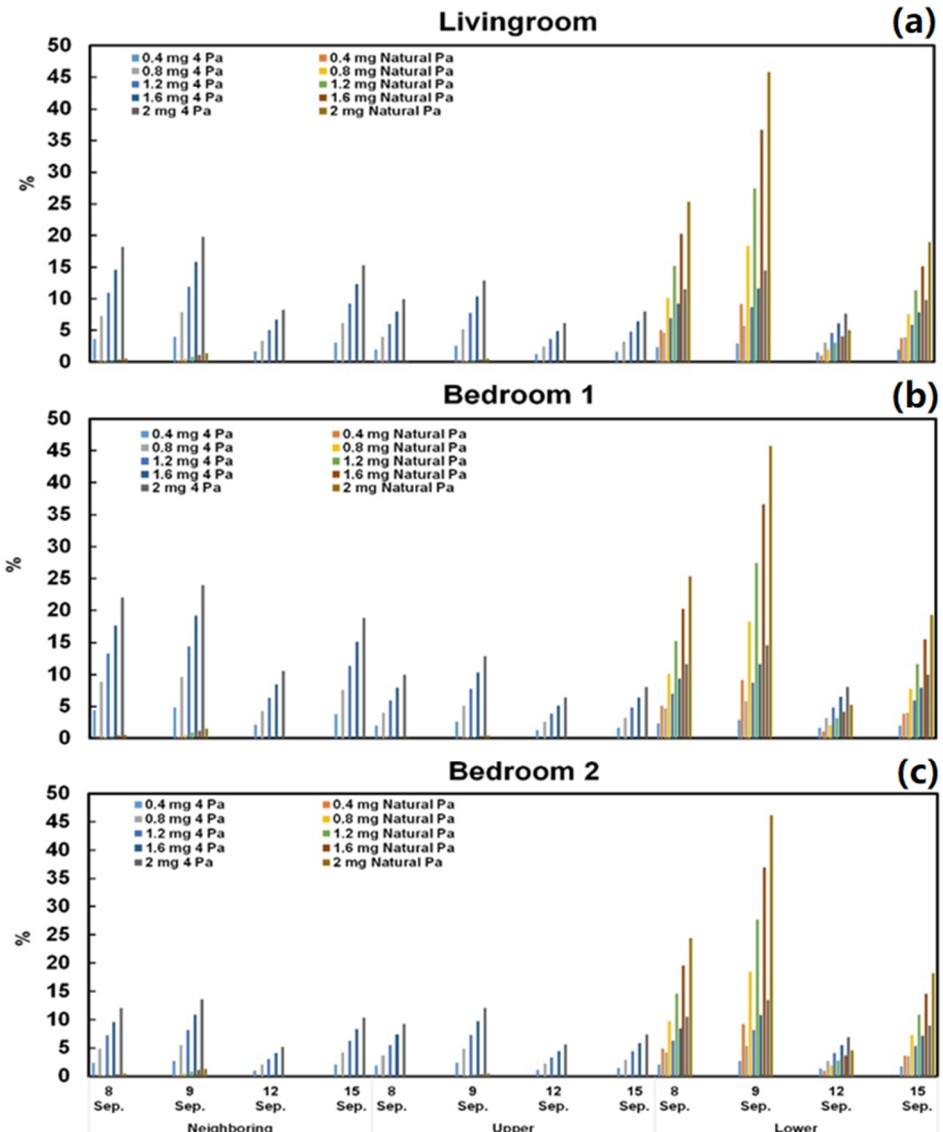

**Figure 8.** The percentage increase of PM$_{2.5}$ concentration in living room (**a**), bedroom 1 (**b**) and bedroom 2 (**c**) in four days (8, 9, 12, and 15 September) with five generation rates (0.4; 0.8; 1.2; 1.6; 2 mg min$^{-1}$) on the three floors (neighboring, upper, and lower apartments) based on two conditions.

Table 4 lists the average estimated penetration factor at 4 Pa pressure difference during two periods (first three-hour and 24-h) for four days. The coefficient of variation (*Cv*) shows a positive trend for the three-hour duration under different emission rates, from around 7% at 0.4 mg min$^{-1}$ to 14% at 2 mg min$^{-1}$ when the sources are in the adjacent apartment. Variation tends to be lower if the sources are on the units above or below. When the blower-door test time extends to 24 h, the variation decreases to less than 2% for all conditions and the estimates close to the input value in CONTAM that calculated from experimental data. It suggests that a short experimental time can result in an obvious error in the blower-door estimates due to the variation of neighboring emissions and weather conditions. Increasing the experimental measurement time could also improve the precision and accuracy regarding the error in different days.

**Table 4.** Average penetration factor under 4 Pa pressure difference during two periods for four days (8, 9, 12, and 15 September) with five generation rates (0.4; 0.8; 1.2; 1.6; 2 mg min$^{-1}$). $Cv$ is the coefficient of variation of the mean.

| Time | Sources Location | | Emission Rate (mg min$^{-1}$) | | | | | | | | |
|---|---|---|---|---|---|---|---|---|---|---|---|
| | | | 0 | 0.4 | 0.8 | 1.2 | 1.6 | 2.0 | Mean | $Cv$ (%) |
| Three hours | Adjacent apartment | Mean | 0.85 | 1.05 | 1.25 | 1.45 | 1.65 | 1.86 | 1.35 | 27.96 |
| | | $Cv$ (%) | 2.95 | 6.94 | 9.28 | 11.42 | 12.95 | 14.01 | | |
| | Upper apartment | Mean | 0.85 | 0.99 | 1.14 | 1.29 | 1.43 | 1.58 | 1.21 | 22.62 |
| | | $Cv$ (%) | 2.95 | 3.62 | 4.21 | 4.94 | 5.01 | 5.59 | | |
| | Lower apartment | Mean | 0.85 | 1.01 | 1.18 | 1.35 | 1.52 | 1.69 | 1.27 | 24.86 |
| | | $Cv$ (%) | 2.95 | 2.84 | 3.04 | 3.22 | 3.31 | 3.43 | | |
| 24 h | Adjacent apartment | Mean | 0.87 | 0.88 | 0.89 | 0.91 | 0.92 | 0.92 | 0.90 | 2.15 |
| | | $Cv$ (%) | 1.44 | 1.43 | 1.41 | 0.64 | 0.63 | 0.54 | | |
| | Upper apartment | Mean | 0.87 | 0.88 | 0.88 | 0.89 | 0.90 | 0.90 | 0.89 | 1.20 |
| | | $Cv$ (%) | 1.44 | 1.61 | 1.43 | 1.59 | 1.07 | 1.28 | | |
| | Lower apartment | Mean | 0.87 | 0.88 | 0.89 | 0.90 | 0.90 | 0.91 | 0.89 | 1.49 |
| | | $Cv$ (%) | 1.44 | 1.43 | 1.08 | 1.44 | 1.28 | 1.27 | | |

### 3.4. A Proposed Experimental Method for Estimating the Distribution of Airflow Paths

It is not always easy to gain long-term access to apartments to conduct extended experiments, so short experiments can bias estimates because of fluctuating sources from neighboring apartments. This means that the precision and accuracy of the blower-door method for estimating PM$_{2.5}$ penetration factors using only outdoor concentration is potentially reduced. In this section, a multi-blower door method is proposed to isolate parts of adjacent areas when estimating air change rate in terms of indoor–outdoor transfer. The idea is to create similar pressure differences between zones so eliminating the air exchange. Strictly speaking, Equations (2) and (3) are only correct in the absence of airflow between inter-unit areas. Considering the effects of neighboring apartments, Equation (2) can be rearranged, assuming the deposition rate is the same for the whole apartment:

$$\frac{dc_i}{dt} = P_o m_o k_{ACH,\Delta p} c_o + P_{ad} m_{ad} k_{ACH,\Delta p} c_{ad} + P_{cor} m_{cor} k_{ACH,\Delta p} c_{cor} - (k_{ACH,\Delta p} + v_{\Delta p}) c_i \quad (7)$$

where $P_o$, $P_{ad}$, $P_{cor}$ are the penetration factors of outside, adjacent units and corridor areas respectively; $m_o$, $m_{ad}$, $m_{cor}$ is air change rate as a proportion of that outside, adjacent units, and corridor, respectively; $c_{ad}$ is the concentration of PM$_{2.5}$ of adjacent units and $c_{cor}$ is the concentration in the corridor. More specifically, the airflow and contaminant transfer from upper, lower, and horizontally positioned adjacent units, can be expressed as:

$$m_{ad} c_{ad} = m_{up} c_{up} + m_{lo} c_{lo} + m_{ho} c_{ho} \quad (8)$$

where $m_{up}$, $m_{lo}$, $m_{ho}$ are air change rate proportions of the upper, lower floor of unit and horizontal adjacent unit respectively; $c_{up}$, $c_{lo}$, $c_{ho}$ the concentration of PM$_{2.5}$ for the upper, lower and horizontal units. Assuming all penetration factors are the same, solving Equation (7) when it is not the steady-state condition by a backward differential scheme, gives indoor PM$_{2.5}$ as a function of time:

$$c_i(t) = P k_{ACH,\Delta p} \left[ m_o c_o(t-1) + m_{up} c_{up}(t-1) + m_{lo} c_{lo}(t-1) + m_{ho} c_{ho}(t-1) + m_{cor} c_{cor}(t-1) \right] \Delta t + \left( 1 - (k_{ACH,\Delta p} + v_{\Delta p}) \Delta t \right) c_i(t-1) \quad (9)$$

Equation (9) combines the relevant factors including upper, lower, horizontal adjacent units, and the external environment under minimum ventilation mode (assuming windows closed, and all holes sealed), and the average indoor concentrations of all zones were assumed to reflect the trend of the whole apartment. As shown in Figure 3, the lift areas of different floors are phantom zones along with the stairwells. Thus, it is hard to create the same pressure difference between stairwell/lift zones and other areas. The gaps and cracks between all zones in the test room and stairwell/lift zones should be sealed because the contribution; those gaps is less than 5%, based on the modeling results. Based on the normal blower-door test, another blower-door fan (Fan 2) is installed in four positions sequentially to isolate gaps between the adjacent zones and test apartment, as shown in Figure 9. Thus, four additional tests should be carried out after measuring the air change rate of the whole test apartment:

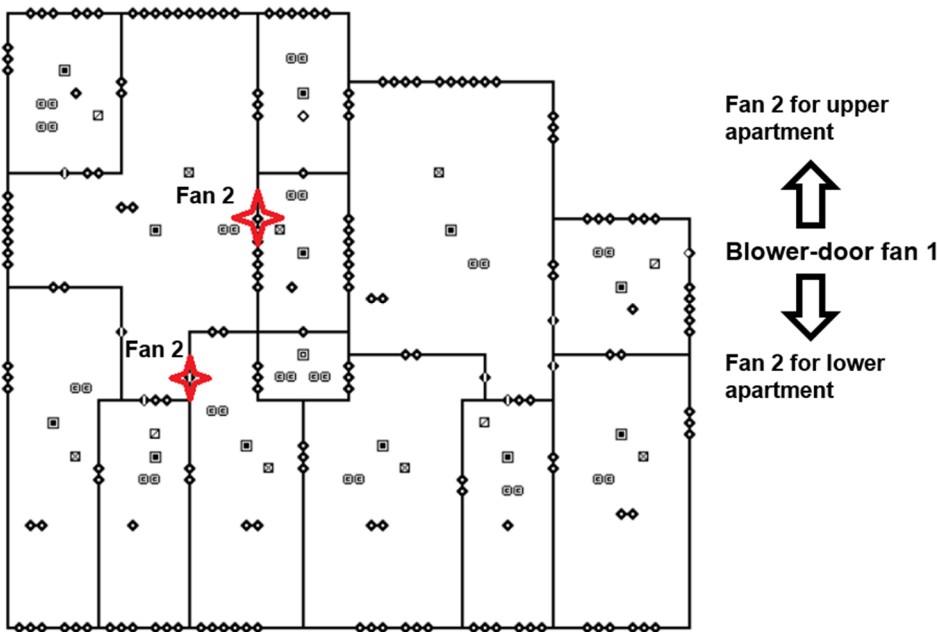

**Figure 9.** The places of Fan 2 for proposed experimental method.

(i) Installing one more blower-door fan (Fan 2) in the same position with the original one in the upper or lower floor apartments. Then setting the same pressure difference between indoors and outdoors for the two apartments and testing the air change rates of two apartments simultaneously. In this case, most air cannot be blown into the test apartment from upper or lower floor areas, thus, the difference in air change rate between two apartments can estimate the air change rate of the connected areas. It should be noted that this consists of two tests, one for the upper floor and another for the lower floor. After this the air change rate from ceiling and floor can be evaluated separately.

(ii) The second fan would then be moved to the door of bedroom 2 of the adjacent apartment and subsequently to the front door of the living room of the neighboring apartment, which connects to the corridor area on the same floor, as shown in Figure 9. Creating the same pressure difference, allows the difference in air change rate to be determined. Thus, the air change rate from corridor to living room and that from the area adjacent to bedroom 1 can be calculated.

Table 5 compares the proportional air change rate of the upper floor, lower floor, corridor, and adjacent area, which were computed by the multi blower-door method of Equation (9) and the CONTAM simulation. Both values show a similar percentage for these areas, with the percentage difference less than 6%. Figure 10 gives an example of the relationship between PM$_{2.5}$ concentrations estimated from Equation (9) and the simulated concentrations at 4 Pa. A slope of 0.99, intercept of 0.13 and $R^2$ of 0.99 were seen, with both

biases only 0.01, suggesting the analytical results have a relatively good agreement with the model. The multi blower-door method is an improvement on the simple blower-door method, reducing the effects of adjacent areas on indoor PM$_{2.5}$ variations. It is believed that using the air change rate from outside only can greatly improve the precision for the blower-door method in estimating the PM$_{2.5}$ penetration factor.

**Table 5.** Air change rate proportion of four adjacent areas and outside.

| Isolated Zone | Air Change Rate Proportion (*m*) | | Absolute Difference | Percentage (%) (100x Absolute Difference/Calculated Value) |
|---|---|---|---|---|
| | Calculated Value | Simulated Value | | |
| Horizontal adjacent apartment ($n_{ho}$) | 0.068 | 0.064 | 0.004 | 6 |
| Upper floor apartment ($n_{up}$) | 0.065 | 0.062 | 0.003 | 5 |
| Lower floor apartment ($n_{lo}$) | 0.053 | 0.056 | 0.003 | 6 |
| Corridor area ($n_{cor}$) | 0.105 | 0.105 | 0.000 | 0 |
| Outside ($n_o$) | 0.709 | 0.713 | 0.003 | 0 |

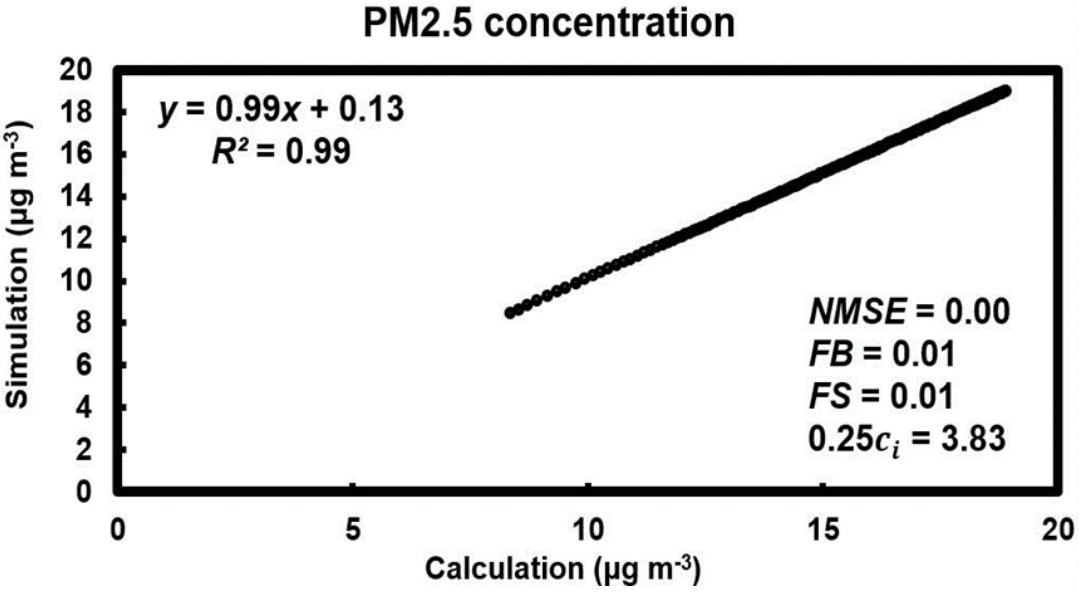

**Figure 10.** An example of the relationship between PM$_{2.5}$ concentrations estimated from Equation 9 and the simulated concentrations at 4 Pa depressurization.

## 4. Conclusions

The variation of indoor PM$_{2.5}$ concentration over time with respect to fluctuating sources in adjacent apartments and different building layouts was simulated in the present work. It predicted indoor PM$_{2.5}$ concentrations under variable weather conditions and investigated the effects of sources from neighboring areas on the accumulation of PM$_{2.5}$ to ascertain the influence of changing environments on estimating particle penetration factors. Both CONTAM simulations and experiments were performed for a residential test building using two methods: the blower-door and traditional decay-rebound method. The results show that the CONTAM modeling has good agreement with the experimental results using both methods, so the model is to be trusted and suitable for exploring scenarios with different sources of PM$_{2.5}$. Assuming outside concentration is not influenced by the emission of neighboring units, a higher emission rate of PM$_{2.5}$ in neighboring apartments always resulted in higher accumulation of indoor PM$_{2.5}$ in the test apartment. When a constant pressure difference was created, the sources in the adjacent apartment on the

same floor had a bigger effect on indoor accumulation, due to the larger air change rate; an important factor that can increase error in the estimates. However, when the decay and rebound method is used, under natural conditions, the emission of PM$_{2.5}$ from neighboring apartments on lower floors significantly effects the concentration in the test apartment because of the stack effect. Although the blower-door method can reduce the variation of deposition rate and penetration factor estimates compared with the natural conditions, the effects of adjacent apartments should be considered. Emission from neighboring units is an important factor that influences the precision and accuracy of the estimated penetration factor during short measuring times. Opening the window in the source apartment and extending the measuring time could reduce the error of the estimates. Simulation suggests that using a multi-blower system to remove ingress from neighboring apartments could also reduce the variation in estimates of penetration factor, which might be more useful for those short time measurements (e.g., less than three hours). The work highlights the effect of transfer of air between apartments on indoor air quality.

**Author Contributions:** Conceptualization of the investigation, methodology, and supervision, I.A.R.; undertaking investigation, preparing data, modeling analysis, and original draft preparation, Y.L.; review editing, P.B. All authors have read and agreed to the published version of the manuscript.

**Funding:** This work was supported by the Research Grants Council of the Hong Kong SAR under Grant number 9042397 (CityU 11334116).

**Acknowledgments:** We thank all of the residents who provided their apartments during this work, particularly all staffs in the Jockey club house in City University of Hong Kong.

**Conflicts of Interest:** The authors declare no conflict of interest.

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
