# Peer review of "Effects of Neighboring Units on the Estimation of Particle Penetration Factor in a Modeled Indoor Environment"

_urbansci, doi:10.3390/urbansci5010002_

Round 1

Reviewer 1 Report

The present study proposes a numerical approach for building infiltration based on CONTAM software. The results obtained were validated against measurements. The available measurements (PM2.5 deposition rate and penetration factor) were made for an apartment of a multistory building using blower-door and decay-rebound methods. The proposed method is considerably interesting and useful for the audience of Urban Science Journal.

The literature review places the study in appropriate context, however, it could be improved with some recent papers in the field. The presentation and discussion of the results and conclusions are satisfactory. However, the quality (resolution) of some Figures must be improved. In general, the manuscript is well written. The “Broad and Specific comments” section should be addressed by authors.

Broad comments

  1. In the “Introduction” Section, I would like to bring to authors’ attention some recent papers using a combination of CONTAM with and without CFD models, which could further improve it.
    1. D. Argyropoulos, H. Hassan, P. Kumar, K.E. Kakosimos, Measurements and modelling of particulate matter building ingress during a severe dust storm event, Build Environ., 167 (2020) 106441.
    2. J. Underhill, W.S. Dols, S.K. Lee, M.P. Fabian, J.I. Levy, Quantifying the impact of housing interventions on indoor air quality and energy consumption using coupled simulation models, Journal of Exposure Science & Environmental Epidemiology, 30 (2020) 436-447.
    3. Zhu, S. Jenkins, K. Addo, M. Heidarinejad, S.A. Romo, A. Layne, J. Ehizibolo, D. Dalgo, N.W. Mattise, F. Hong, O.O. Adenaiye, J.P. Bueno de Mesquita, B.J. Albert, R. Washington-Lewis, J. German, S. Tai, S. Youssefi, D.K. Milton, J. Srebric, Ventilation and laboratory confirmed acute respiratory infection (ARI) rates in college residence halls in College Park, Maryland, Environ. Int., 137 (2020) 105537
  2. It is known that the CONTAM uses average wind pressure coefficients for a simplified building geometries according to the relationships of Swami and Chandra (1987). Why did you not use a CFD approach for calculating the wind pressure coefficients or the CFD tool of CONTAM (complex building)? Please also mention the version of CONTAM that you used for the current simulations.

Specific comments

    1. Page 3, line 148: “wereadopted”, split the two words
    2. Figures 4 and 5, increase the resolution.

Author Response

THANK YOU FOR THESE USEFUL THOUGHTS- RESPONSES AT THE END

Reviewer 1

The present study proposes a numerical approach for building infiltration based on CONTAM software. The results obtained were validated against measurements. The available measurements (PM2.5 deposition rate and penetration factor) were made for an apartment of a multistory building using blower-door and decay-rebound methods. The proposed method is considerably interesting and useful for the audience of Urban Science Journal.

The literature review places the study in appropriate context, however, it could be improved with some recent papers in the field. The presentation and discussion of the results and conclusions are satisfactory. However, the quality (resolution) of some Figures must be improved. In general, the manuscript is well written. The “Broad and Specific comments” section should be addressed by authors.

Broad comments

In the “Introduction” Section, I would like to bring to authors’ attention some recent papers using a combination of CONTAM with and without CFD models, which could further improve it.

  1. Argyropoulos, H. Hassan, P. Kumar, K.E. Kakosimos, Measurements and modelling of particulate matter building ingress during a severe dust storm event, Build Environ., 167 (2020) 106441.
  2. Underhill, W.S. Dols, S.K. Lee, M.P. Fabian, J.I. Levy, Quantifying the impact of housing interventions on indoor air quality and energy consumption using coupled simulation models, Journal of Exposure Science & Environmental Epidemiology, 30 (2020) 436-447.

Zhu, S. Jenkins, K. Addo, M. Heidarinejad, S.A. Romo, A. Layne, J. Ehizibolo, D. Dalgo, N.W. Mattise, F. Hong, O.O. Adenaiye, J.P. Bueno de Mesquita, B.J. Albert, R. Washington-Lewis, J. German, S. Tai, S. Youssefi, D.K. Milton, J. Srebric, Ventilation and laboratory confirmed acute respiratory infection (ARI) rates in college residence halls in College Park, Maryland, Environ. Int., 137 (2020) 105537

It is known that the CONTAM uses average wind pressure coefficients for a simplified building geometries according to the relationships of Swami and Chandra (1987). Why did you not use a CFD approach for calculating the wind pressure coefficients or the CFD tool of CONTAM (complex building)? Please also mention the version of CONTAM that you used for the current simulations.

ADDED THE SUMMARY OF THE REFERENCES IN THE INTRODUCTION.

WHEN CONSIDERING THE EFFECTS OF VARIABLE EXTERNAL WIND PRESSURES IN A NON-CUBICAL SURFACE, THE CFD MODULE IS PROPOSED TO USE FOR ACHIEVING A HIGHER ACCURATE ESTIMATION OF WIND PRESSURE. HOWEVER, THE MULTI-ZONE CONTAM MODEL WITHOUT CFD MIGHT BE BETTER IF WE FOCUS ON THE WHOLE-BUILDING AND YEARLY DYNAMIC SIMULATIONS, MODELING BUILDING AIR INFILTRATION AND COMPUTATIONAL SPEED. WHEN THE BLOWER-DOOR METHOD WAS USED, THE EFFECTS OF INTERIOR PARTITION SHOULD BE NEGLIGIBLE AND THE WHOLE BUILDING AVERAGE INFILTRATION WAS ESTIMATED. THE DIRECTION OF EXTERNAL AIR PRESSURE CAN BE CONTROLLED DUE TO THE STEADY-STATE INDOOR-OUTDOOR PRESSURE DIFFERENCE FOR REDUCING THE EFFECTS OF VARIABLE EXTERNAL WIND PRESSURE, AND THE ESTIMATES OF DEPOSITION RATE AND PENETRATION FACTOR WERE CALCULATED BY INDOOR AIR QUALITY MODEL. THUS, THE AVERAGE WIND PRESSURE PROFILE FOR CONTAM BASED ON MULTI-ZONE MIGHT BE BETTER CONSIDERING THE BLOWER-DOOR METHOD. WE ALSO MENTIONED THAT WE USED THE CONTAM VERSION 3.2 (NIST, GAITHERSBURG, MD, USA) FOR MODELING.

Specific comments

Page 3, line 148: “wereadopted”, split the two words

DONE IT.

Figures 4 and 5, increase the resolution.

DONE IT.

Reviewer 2 Report

As a reviewer I have the following remarks.

  1. Abstract. Line 14: “important source of fine particulate matter (PM2.5)”.
  2. I think a reader needs an explanation. I used the Internet: “CONTAM is a multizone indoor air quality and ventilation analysis computer program designed to help you determine”.
  3. Line 94. Is it possible to provide height also height of a room. “One apartment on the fifth floor “ – meters above the ground?
  4. Line 100. “The doors of the corridor connecting the stairway were normally closed” – I assume that is a lift. An impact of it?
  5. Line 141, Pleas consider the following (4, 6, 8, 10, 12, 14, and 16 Pa).
  6. Figure 2. It looks strange. Any reason not to start with x-axis, say from 6? In general, in figure presentation we don’t need to show so much of empty area.
  7. Line 151: is it correct? “assumed the interior well-mixed without indoor sources”, as in the formula you have outdoor?
  8. Line 161. Be more precise: “the time-series indoor particle concentration”.
  9. Line 161: “solved by backward differential” or “difference”?
  10. Table 1. We don’t know the meaning of SD, and also 30.36 (0.32), what is it (0.32) – please add notation.
  11. Formulas 4-6, why we need a bar under Cp and Co?
  12. Line 269. “simulation should under the range from 0.75” – I think you need a verb, check also other sentences.
  13. Line 281. You have Fig.4a, 4c, 4e and 4g, consider Fig. 4 (a, c, e, and g).
  14. Line 288: “on 11 Sep”, Sep? Ugly.
  15. Line 307. in Livingroom.
  16. Line 315: “March, Jun, September and December” = June.
  17. Line 443: “(8 Sep.; 9 Sep.; 12 Sep; 15 Sep.)” – I suggest: (8, 9, 12, and 15 September). Check/adjust other situations.
  18. I suggest to check the paper carefully, using my remarks, many on notations/abbreviations.

Thank you

Author Response

THANK YOU FOR THESE USEFUL THOUGHTS- RESPONSES AT THE END

Abstract. Line 14: “important source of fine particulate matter (PM2.5)”.

I think a reader needs an explanation. I used the Internet: “CONTAM is a multizone indoor air quality and ventilation analysis computer program to aid the prediction of indoor air quality.”.

AGREE WITH IT. I ADDED THE EXPLANATION IN THE ABSTRACT.

Line 94. Is it possible to provide height also height of a room. “One apartment on the fifth floor “ – meters above the ground?

YES, AROUND 15.6 M ABOVE THE GROUND. DONE It.

Line 100. “The doors of the corridor connecting the stairway were normally closed” – I assume that is a lift. An impact of it?

CHANGED TO “The doors of the corridor connecting the stairway were normally closed as the door is used as an emergency exit. The lift door was also closed during the test to reduce the stack effect in the building.”

Line 141, Pleas consider the following (4, 6, 8, 10, 12, 14, and 16 Pa).

DONE IT.

Figure 2. It looks strange. Any reason not to start with x-axis, say from 6? In general, in figure presentation we don’t need to show so much of empty area.

DONE IT.

Line 151: is it correct? “assumed the interior well-mixed without indoor sources”, as in the formula you have outdoor?

YES,  AND  ARE THE AVERAGE INDOOR AND OUTDOOR PARTICLE CONCENTRATIONS, RESPECTIVELY IN THE EQUATION 2.

Line 161. Be more precise: “the time-series indoor particle concentration”.

Line 161: “solved by backward differential” or “difference”?

CHANGED TO “the indoor particle concentration can be solved by a backward differential scheme for a given time step, as shown below:.

Table 1. We don’t know the meaning of SD, and also 30.36 (0.32), what is it (0.32) – please add notation.

DONE IT.

Formulas 4-6, why we need a bar under Cp and Co?

THE TOP BAR REPRESENTS THE AVERAGE VALUE NOW MENTIONED IN TEXT.

Line 269. “simulation should under the range from 0.75” – I think you need a verb, check also other sentences.

CHANGED TO “The regression line between measurements and simulations should have a slope (M) between 0.75 and 1.25.”

Line 281. You have Fig.4a, 4c, 4e and 4g, consider Fig. 4 (a, c, e, and g).

DONE IT.

Line 288: “on 11 Sep”, Sep? Ugly.

SEP. REPRESENTS THE SEPTEMBER. ALL CHANGED TO SEPTEMBER.

Line 307. in Livingroom.

DONE IT.

Line 315: “March, Jun, September and December” = June.

DONE IT.

Line 443: “(8 Sep.; 9 Sep.; 12 Sep; 15 Sep.)” – I suggest: (8, 9, 12, and 15 September). Check/adjust other situations.

DONE IT.

I suggest to check the paper carefully, using my remarks, many on notations/abbreviations.